# Intrinsic Lipid Curvature and Bilayer Elasticity as Regulators of Channel Function: A Comparative Single-Molecule Study

**DOI:** 10.3390/ijms25052758

**Published:** 2024-02-27

**Authors:** Mohammad Ashrafuzzaman, Roger E. Koeppe, Olaf S. Andersen

**Affiliations:** 1Department of Physiology and Biophysics, Weill Cornell Medicine, New York, NY 10065, USA; mzzaman@ksu.edu.sa; 2Department of Chemistry and Biochemistry, University of Arkansas, Fayetteville, AR 72701, USA; rk2@uark.edu

**Keywords:** lipid intrinsic curvature, elasticity, amphiphiles, bilayer-mediated regulation, gramicidin channel, alamethicin channel

## Abstract

Perturbations in bilayer material properties (thickness, lipid intrinsic curvature and elastic moduli) modulate the free energy difference between different membrane protein conformations, thereby leading to changes in the conformational preferences of bilayer-spanning proteins. To further explore the relative importance of curvature and elasticity in determining the changes in bilayer properties that underlie the modulation of channel function, we investigated how the micelle-forming amphiphiles Triton X-100, reduced Triton X-100 and the H_II_ lipid phase promoter capsaicin modulate the function of alamethicin and gramicidin channels. Whether the amphiphile-induced changes in intrinsic curvature were negative or positive, amphiphile addition increased gramicidin channel appearance rates and lifetimes and stabilized the higher conductance states in alamethicin channels. When the intrinsic curvature was modulated by altering phospholipid head group interactions, however, maneuvers that promote a negative-going curvature stabilized the higher conductance states in alamethicin channels but destabilized gramicidin channels. Using gramicidin channels of different lengths to probe for changes in bilayer elasticity, we found that amphiphile adsorption increases bilayer elasticity, whereas altering head group interactions does not. We draw the following conclusions: first, confirming previous studies, both alamethicin and gramicidin channels are modulated by changes in lipid bilayer material properties, the changes occurring in parallel yet differing dependent on the property that is being changed; second, isolated, negative-going changes in curvature stabilize the higher current levels in alamethicin channels and destabilize gramicidin channels; third, increases in bilayer elasticity stabilize the higher current levels in alamethicin channels and stabilize gramicidin channels; and fourth, the energetic consequences of changes in elasticity tend to dominate over changes in curvature.

## 1. Introduction

Membrane protein function is regulated by changes in lipid bilayer composition [1,2,3,4,5,6,7,8,9,10,11,12]. This regulation is, in part, due to changes in membrane physical properties, including thickness (*d*_0_), intrinsic lipid curvature (*c*_0_), bilayer compression (*K*_a_) and bending (*K*_c_) moduli, e.g., [8]. The bilayer regulation of membrane protein function occurs because, first, membrane proteins undergo conformational changes; second, hydrophobic interactions between lipid bilayers and their embedded proteins cause bilayers to adapt to the proteins’ hydrophobic exterior (and vice versa); and third, the hydrophobic adaptation between bilayers and proteins causes protein conformation changes to alter the packing in the adjacent bilayer. This bilayer deformation will incur an energetic cost that contributes to the total free energy difference between two protein conformations (functional states), I and II ($\Delta G_{\mathrm{tot}}^{I\to \mathrm{II}}$), [8,13,14]:(1)$$\Delta G_{\mathrm{tot}}^{I\to \mathrm{II}}=\Delta G_{\mathrm{prot}}^{I\to \mathrm{II}}+\Delta G_{\mathrm{bil}}^{I\to \mathrm{II}}$$
where $\Delta G_{\mathrm{prot}}^{I\to \mathrm{II}}$ denotes the energetic cost of the rearrangements within the protein that underlie the protein conformational change and $\Delta G_{\mathrm{bil}}^{I\to \mathrm{II}}$ the bilayer contribution to $\Delta G_{\mathrm{tot}}^{I\to \mathrm{II}}$ ($\Delta G_{\mathrm{bil}}^{I\to \mathrm{II}}=\Delta G_{\mathrm{def}}^{\mathrm{II}}-\Delta G_{\mathrm{def}}^{I}$, where Δ*G*_def_ denotes the energetic cost of the bilayer deformation (changes in organization and dynamics of the lipid molecules adjacent to the protein) caused by the hydrophobic adaptation to each protein conformation). Δ*G*_def_ varies as a function of *d*_0_, *c*_0_, *K*_a_ and *K*_c_, and the protein’s shape, including the hydrophobic length (*l*) [6,14,15], changes in any of these will alter $\Delta G_{\mathrm{bil}}^{I\to \mathrm{II}}$ and, thus, protein function.

Studies on purified proteins reconstituted in lipid bilayers of defined composition show that membrane protein function is altered by experimental maneuvers that alter bilayer thickness or lipid intrinsic curvature; for a review, see [6]. (In addition to changes in intrinsic curvature, membrane protein organization and function may also be modulated by changes in overall membrane curvature [16,17] in which the two interfaces have opposite curvatures; we will not consider such changes). It is generally accepted that the bilayer thickness-dependent changes in membrane protein function result from changes in bilayer thickness per se. It remains unclear to what extent changes in intrinsic lipid curvature per se alter membrane protein function because experimental manipulations that alter the curvature, e.g., replacing phosphatidylcholine by phosphatidylethanolamine head groups, also alter other bilayer properties, such as bilayer thickness [18,19,20,21] and the ability to form hydrogen bonds [22,23]. Moreover, amphiphiles that cause positive or negative changes in intrinsic lipid curvature, e.g., Triton X-100 or capsaicin, have similar effects on the function of gramicidin and voltage-gated sodium channels [24], indicating that they alter bilayer properties other than curvature—e.g., lipid bilayer elasticity [14,25,26,27,28]—and that the (amphiphile-induced) curvature changes are less important determinants of channel function in these experiments.

To further explore the relative importance of curvature and other bilayer properties, such as elasticity, we explore how molecules that produce positive curvature (Triton X-100, TX100 and reduced Triton X-100, rTX100) and negative curvature changes (capsaicin, Cpsn) alter the function of alamethicin channels and relate the changes in alamethicin channel function to the changes in gramicidin channel function. These channels are regulated by maneuvers that alter intrinsic lipid curvature [21,29,30] in hydrocarbon-containing planar lipid bilayers where thickness changes are minimal [21,30,31], making them suitable for the present study.

The alamethicins and the linear gramicidins are channel-forming peptide antibiotics that are synthesized by nonribosomal peptide synthases (NRPs) [32]. As for most peptides synthesized by NRPs, the sequences have nongenetic amino acids: in the case of the gramicidins, D-amino acids; in the case of alamethicin, α-aminoisobutyric acid (Aib) and phenylalaninol (Pheol). The linear gramicidins are 15 amino acid peptides produced by the soil bacillus *Bacillus brevis*. They occur in a number of sequence variants; the predominant species, [Val^1^]gramicidin A (or gA), has the sequence listed in Figure 1A. The alamethicins are 20 amino acid peptides produced by the soil fungus *Trichoderma viride*. As is the case for the linear gramicidins, alamethicin occurs in a number of sequence variants; the predominant species, alamethicin I (or Alm), has the sequence listed in Figure 1B.

gA and Alm channels have been used extensively to probe changes in lipid bilayer properties as sensed by bilayer-spanning channels; for reviews, see [8,33]. They complement each other because of the different channel structures and mechanisms of formation (Figure 1).

**Figure 1 ijms-25-02758-f001:**
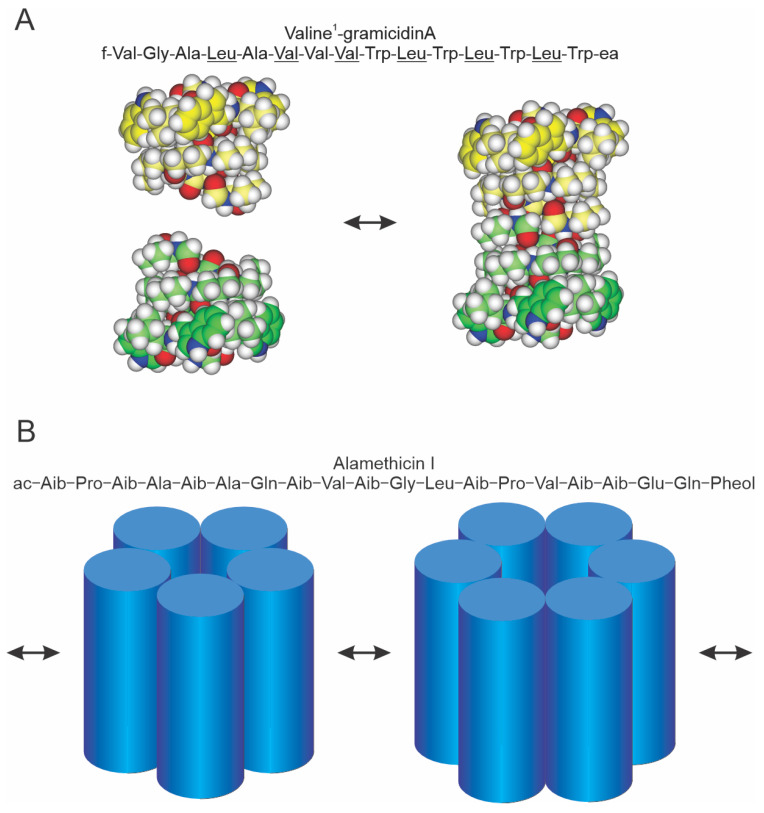
Schematic models of gramicidin and alamethicin channels. (**A**) Top: sequence of [Val^1^]gA [34], the major gramicidin species in naturally occurring mixture of peptides [35]; f is formyl, ea ethanolamine and the D-amino acids are underlined. Bottom: gramicidin channels form and disappear, as indicated by the arrows, by a transmembrane association/dissociation [36]. Left, atomic resolution structures of the β^6.3^-helical monomers, the two subunits are depicted some distance apart; right, atomic resolution structure of the β^6.3^-helical conducting dimer. The carbons in the two subunits are colored green and yellow, respectively, with the carbon atoms in the Trp side chains emphasized. Blue is nitrogen, red is oxygen and white is hydrogen. (**B**) Top: sequence of alamethicin I [37], the major species of alamethicin; ac is acetate, Aib α-isobutyric acid and Pheol phenylalcohol. Bottom: different interconverting oligomeric states, as indicated by the arrows, of the bilayer-spanning channel. The number of subunits may change by the association/dissociation of bilayer-spanning subunits or oligomers or by the accretion of subunits at the bilayer/solution interface that inserts into the bilayer [33,38].

gA channels (Figure 1A) form by the bimolecular association of two nonconducting subunits [39,40,41,42,43], one from each bilayer leaflet [36]; see [44] for a summary of the evidence. The channel structure, a β^6.3^-helical dimer, is known from high-resolution solution nuclear magnetic resonance (NMR) studies on gA incorporated into sodium dodecyl sulfate micelles [45,46] and solid-state NMR studies on gA incorporated into oriented lipid bilayers [47,48]. Minor differences between the structures determined in solution NMR and solid-state NMR can be resolved using molecular dynamics simulations to allow for local motion and motional averaging [49].

Except for extreme changes in bilayer thickness [50,51,52] or curvature [53], the gramicidin channel structure is remarkably unaffected by changes in bilayer thickness [54,55]. The current transitions associated with channel formation and disappearance have a single predominant amplitude [56] that varies little with changes in bilayer thickness [27,57,58].

The channel’s hydrophobic length is less than the bilayer hydrophobic thickness, meaning that channel formation leads to a local bilayer thinning [59,60,61,62]. The free energy difference of channel formation ($\Delta G_{\mathrm{tot}}^{M\to D}$), thus, will include a contribution from the lipid bilayer ($\Delta G_{\mathrm{bil}}^{M\to D}$), and the gA monomer↔dimer equilibrium will vary as a function of changes in lipid bilayer properties [8]. These gA channel characteristics make them useful as tools to probe changes in lipid bilayer properties, as sensed by bilayer-spanning proteins.

Less is known about Alm channels, which form voltage-dependent multi-state channels [63,64]; for reviews, see [33,38,65,66,67,68]. Alm was crystallized from organic solvents, and the structure was solved by X-ray crystallography [69], which revealed a predominantly α-helical structure. The structure of the bilayer-associated Alm is α-helical, as deduced from solid-state NMR [70] and oriented circular dichroism spectroscopy [71]. The orientation of the bilayer-associated Alm varies as a function of the Alm/lipid mole fraction [38,71,72]. At low Alm/lipid mole fractions (<1/100), where Alm is monomeric [70], the helical axis is parallel with the bilayer/solution interface; at high Alm/lipid mole fractions (>1/50), the helical axis is perpendicular to the bilayer/solution interface. The transition between these two states depends on the bilayer composition [38] and occurs at Alm/lipid ratios near the cooperative transition in the adsorption isotherm [73]. The switch between the parallel (adsorbed) and perpendicular (inserted) state can be understood in terms of a build-up of elastic curvature stress in the bilayer [74], as the bilayer thickness decreases with increasing Alm mole fraction until the mole fraction where the switch from the adsorbed to inserted state occurs [38].

Alm channels are barrel-stave assemblies of bilayer-spanning α-helices [75,76,77], as originally proposed by Bauman and Mueller [78] and Boheim [64], with multiple current levels. It is generally accepted that the different conductance states reflect different peptide stoichiometries. It remains unclear whether the different current levels represent transitions within a single multimeric barrel-stave channel with different Alm stoichiometries [64,78,79] or an array of closely packed parallel pores [80], where the different current levels reflect the association/dissociation of Alm monomers or nonconducting Alm aggregates. Neutron [81] and X-ray diffraction [76] experiments provide support for a well-defined water-filled pore with a 10° to 20° tilt relative to the bilayer normal as deduced by solid-state NMR [82]. Probing the pore diameter using polyethyleneglycols of different sizes [80] provides support for a multipore cluster. Both models are consistent with mechanoelectrical experiments [79], which show that the different conductance states differ in total area by ~1.2 nm^2^—a number that can be decomposed into a contribution from the Alm monomer, ~0.8 nm^2^ [83], and from the water-filled pore, ~0.4 nm^2^, with the latter being similar to the area changes deduced from the measured current changes [84].

Both gA and Alm channels are sensitive to changes in bilayer properties. In the case of Alm channels, the higher current levels are stabilized by changes in phospholipid head groups and head group interactions that cause negative changes in intrinsic lipid curvature [29,33,85]. In the case of gA channels, the single-channel appearance rates and lifetimes (and, thus, the time-averaged channel surface densities and the channel activity) are reduced by changes in head group interactions that cause negative changes in curvature [21,30]. Yet, the role of curvature in the regulation of gA channel function is complex because channel activity is increased by reversibly adsorbing amphiphiles that produce positive as well as negative changes in curvature [24,27]. These seemingly contradictory results arise because reversibly adsorbing amphiphiles for thermodynamic reasons [25,26] increases lipid bilayer elasticity [14,24,25,26,28,86,87,88].

A similar result was observed with voltage- and ligand-gated channels in biological membranes. Reversibly adsorbing amphiphiles inhibit voltage-gated sodium channels (Na_V_) by causing a hyperpolarization shift in the steady-state channel availability curve [24,89] and causing the desensitization of the GABA_A_ receptor [90]. In either case, amphiphiles that cause positive and negative changes in curvature cause similar changes in channel function, and, in the case of Na_V_, the shift in inactivation is correlated with the changes in gA channel lifetime [14,24].

We, therefore, explored how the reversibly partitioning amphiphiles TX100, rTX100 and Cpsn alter the function of Alm channels and related the changes in the Alm channel function to the changes in gA channel lifetimes. We find that the amphiphile-induced changes in the Alm channel function (the distribution among the different current levels) are correlated to the changes in the gA channel function (single-channel lifetime). We conclude that changes in bilayer elasticity are more important than changes in curvature in terms of modulating Alm channel function, like what was found with gA channels.

## 2. Results

### 2.1. Amphiphiles Modulate Alm Channel Function

TX100, rTX100 and Cpsn are potent modifiers of Alm channel function. Figure 2 shows current traces obtained before and after the addition of Cpsn, TX100 or rTX100 to both sides of a bilayer (Alm was present on one side only; potentials are measured relative to the Alm-free solution). Amphiphiles may exert their effects by mechanisms that do not involve changes in bilayer mechanical properties, including changes in the interfacial dipole potential [91,92], and asymmetric addition of amphiphiles that alter the dipole and/or surface potential may exert their effects on Alm channel function by changes in the electric field within the membrane [93], a complication we strived to avoid with symmetric addition of the amphiphiles. Because we wished to focus on the amphiphiles’ bilayer effects, all experiments were performed at a single membrane potential (150 mV).

The three amphiphiles increased the Alm channel activity (time-averaged number of conducting channels) and shifted the distribution among current levels toward higher current levels. The short time traces at the bottom of Figure 2 show that the different conduction levels in Alm channels do not vary in the presence of TX100, rTX100 and Cpsn. (The current levels observed in the absence of amphiphiles are summarized in Appendix A; Table 1 summarizes information on the lack of amphiphile effects on the current levels). TX100 and rTX100 were equally effective in producing the change in Alm channel function; three-fold higher concentrations of Cpsn were needed to observe changes comparable to those observed with TX100 and rTX100. Though the amphiphiles did not cause obvious changes in the baseline current, the bilayers were destabilized. In the absence of amphiphiles, we could record current traces for several minutes; in the presence of amphiphiles, we usually could record for no more than 2 min before the small membrane (patch) broke.

Alm channels occur as bursts of activity (Figure 2), and the time spent in the different Alm channel conducting states, as well as the time when no Alm channel activity could be observed, was determined from all-point histograms [94] based on 1–3 min current traces (Figure 3), which showed that the amphiphiles increased the channel activity and shifted the distribution of current levels toward higher levels.

For each histogram, the peaks representing the no-channel state (current level *nc*) and the different conducting states (current levels 0, 1, 2, 3 … *n*) were identified and fitted by Gaussian fits, and the area under each peak was calculated to determine the time spent in that channel state. We sometimes observed multiple Alm channels at the same time, which complicated the assignment of the peaks in the histograms to channel states. We tried to exclude such multiple channel records from the analysis, which may lead to an underestimation of the amphiphile-induced changes in Alm channel activity. This was not always possible after the addition of Cpsn, TX100 or rTX100, which increased channel activity. In this case, we assigned the peaks in the all-points histograms to the underlying current levels. When that was not possible, we did not use the data.

The reported results are based on at least 60 s of continuous recording to minimize the errors introduced by temporal changes in channel activity. On a given day, it was usually possible to obtain three recordings in three different small membranes; each data point is based on at least recordings obtained on at least three different days.

To facilitate comparison of different experiments, each histogram was adjusted (by a few pA) such that the peak representing the baseline was at 0 pA. In both the absence and presence of TX100, each current level is indicated by a well-defined peak, which does not vary as the amphiphile is added (similar results were found with rTX100 and Cpsn). Following Hanke and Boheim [84], the current levels are denoted as 0, 1, 2, 3, etc., with the highest level we observe in the presence of the amphiphiles being 7. Table 1 summarizes our results for TX100, rTX100 and Cpsn. The amphiphiles do not shift the current levels, indicating that the amphiphiles produce minimal changes in the channel structure associated with the different current levels. This result is in general agreement with a barrel-stave structure for the conducting channels [76,81].

In some cases, there were small, secondary peaks between the main current levels (e.g., Figure 3, histograms in right column). These peaks represent the summed current through two different channels, e.g., level 0 + level 0, denoted as (0 + 0); level 1 + level 0, denoted as (1 + 0). Though we strived to use records with only single channels, these secondary peaks often occurred after amphiphile addition because the amphiphiles increased the time-averaged number of conducting channels; they were incorporated into the data analysis and final results in case we could unambiguously assign the peaks to different current level combinations. If that was not possible, the results were not used.

The Alm channel activity was quantified as the ratio of time spent in any conducting level relative to the *nc* level (*R*_Alm_):(2)$$R_{\mathrm{Alm}}=\left(\sum_{i=0}^{n}A_{i}\right)/A_{\mathrm{nc}};$$
where *A*_nc_ denotes the area under the peak representing the baseline current (the no-channel state) and *A*_i_ the area under the peak representing current level *i* (*i* = 0, 1, 2, 3, …, *n*, using the nomenclature of [84], where *n* + 1 is the maximal number of current levels observed in the experiment).

The amphiphile-induced changes in Alm channel activity could, thus, be quantified as
(3)$$\frac{R_{\mathrm{Alm}}^{\mathrm{AM}}}{R_{\mathrm{Alm}}^{\mathrm{cntl}}}=\frac{\left(\sum_{i=0}^{n^{\prime }}A_{i}^{\mathrm{AM}}\right)/A_{\mathrm{nc}}^{\mathrm{AM}}}{\left(\sum_{i=0}^{n}A_{i}^{\mathrm{cntl}}\right)/A_{\mathrm{nc}}^{\mathrm{cntl}}},$$
where the superscripts AM and cntl denote results obtained in the presence (AM) or absence (cntl) of amphiphile, and *n* + 1 and *n*′ + 1 denote the maximum number of current levels (including level 0) observed in the absence and presence of the amphiphile.

The overall channel activity, evaluated as *R*_Alm_ (Equation (2)), is stationary over the duration of the recordings. Figure 4 shows the variation in *R*_Alm_ evaluated over 10 s time intervals as a function of time.

The Alm channel activity varied over time (as also evident in the traces in Figure 2), but there was no systematic trend.

The probability of a channel being in current level *k* (*W*_k_) was estimated as follows:(4)$$W_{k}=\frac{A_{k}}{A_{\mathrm{nc}}+\sum_{i=0}^{n}A_{i}}$$
where *A*_nc_ and *A*_i_ *i* (*i* = 0, 1, 2, 3, …, *n*; *n* + 1 is the maximal number of current levels observed in the experiment) denote the areas under the peak representing current level, which is proportional to the time spent in each current level. (The nominator in Equation (4) is the total area under the histogram, which is proportional to the recording time for that experiment).

The time spent in current level *k*, relative to the no-channel state (*nc*), *A*_k_/*A*_nc_, provides a measure of ∆*G*^nc→k^, the free energy difference current level *k* relative to the *nc* state:(5)$$\Delta G^{\mathrm{nc}\to k}=k_{B}T\cdot \ln\{W_{k}/W_{\mathrm{nc}}\}=k_{B}T\cdot \ln\{A_{k}/A_{\mathrm{nc}}\},$$
where *T* is the temperature in kelvin and *k*_B_ Boltzmann’s constant. The amphiphile-induced changes in channel function (the ratio of the probabilities of being in current level *k* relative to the *nc* level) can then be expressed as follows:(6)$$\frac{A_{k}^{\mathrm{AM}}/A_{\mathrm{nc}}^{\mathrm{AM}}}{A_{k}^{\mathrm{cntl}}/A_{\mathrm{nc}}^{\mathrm{cntl}}}=\exp\left\{-\frac{\Delta G^{\mathrm{nc}\to k,\mathrm{AM}}-\Delta G^{\mathrm{nc}\to k,\mathrm{cntl}}}{k_{B}T}\right\}=\exp\left\{-\frac{\Delta \Delta G^{\mathrm{nc}\to k}}{k_{B}T}\right\}.$$

The distribution among current levels, e.g., between levels *j* and *k*, was determined from the amphiphile-induced changes in *A*_k_/*A*_j_; it was, thus, possible to obtain estimates for the free energy differences between the two channel states:(7)$$\frac{A_{k}^{\mathrm{AM}}/A_{j}^{\mathrm{AM}}}{A_{k}^{\mathrm{cntl}}/A_{j}^{\mathrm{cntl}}}=\exp\left\{-\frac{\Delta G^{j\to k,\mathrm{AM}}-\Delta G^{j\to k,\mathrm{cntl}}}{k_{B}T}\right\}=\exp\left\{-\frac{\Delta \Delta G^{j\to k}}{k_{B}T}\right\}.$$

The amphiphile-induced changes in Alm channel activity are summarized in Figure 5, Figure 6 and Figure 7. Figure 5 shows the changes in overall channel activity, *R*_Alm_, which increases as an approximately linear function of [AM]. Figure 6 shows the changes in the distribution of current levels relative to the no-channel level (cf. Equation (5)), where the results are plotted as $\ln\left\{\left(A_{i}^{\mathrm{AM}}/A_{\mathrm{nc}}^{\mathrm{AM}}\right)/\left(A_{i}^{\mathrm{cntl}}/A_{\mathrm{nc}}^{\mathrm{cntl}}\right)\right\}$ (=−∆∆*G*^nc→i^*/k*_B_*T*) vs. [AM]. Figure 7 shows the changes in the stability of current levels 2 and 3 relative to level 1 (cf. Equation (7)). The amphiphile-induced stabilization is more pronounced for the higher current levels (Figure 6 and Figure 7), with a tendency to level off at higher [AM].

The amphiphile-induced changes in Alm channel function, with amphiphiles that promote negative and positive curvatures having similar effects, were surprising in light of the results of Keller et al. [29] and Bezrukov et al. [85], who showed that changes in phospholipid head group size [29] or charge [85] that promote negative curvatures stabilize the higher Alm channel current levels. Given the slightly different membrane compositions, Keller et al. [29] and Bezrukov et al. [85] used *n*-hexadecane, and we used *n*-decane, so we reexamined whether this relation also holds in our system. It did, as shown in Appendix A, confirming that Alm channels are, indeed, regulated by changes in curvature. We conclude that changes in bilayer properties other than curvature are also important for Alm channel function, which we explored using the gA channels as probes.

### 2.2. Amphiphile Modulation of gA Channel Stability

TX100, rTX100 and Cpsn are potent modifiers of gA channel function [24,89,95]. Figure 8 shows single-channel current traces before and after the addition of 10 µM TX100 or 30 µM Cpsn to the aqueous phases bathing a DOPC bilayer doped with gA^−^(13) and AgA(15).

These experiments were analyzed by constructing single-channel current transition amplitude histograms [96], where the current transitions associated with each channel type appear as a single peak, which allows for determining the single-channel lifetimes (τ) for each channel type [13] by matching channel appearances to disappearances, e.g., [97]; lifetime histograms were constructed and transformed into survivor distributions. The average channel lifetimes (τ = 1/*k*_−1_, where *k*_−1_ is the dissociation rate constant for the bilayer-spanning, conducting channel, D) were determined by fitting a single exponential distribution, *N*(*t*)/*N*(0) = exp{−*t*/τ}, where *N*(*t*) denotes the number of channels with a lifetime longer than time *t*, to the normalized survivor distributions.

The dimeric gA channels (D) form by the transbilayer association of two non-conducting monomers (M):$$2\;M\overset{\;k_{1}\;}{\underset{\;k_{-1}\;}{⇄}}D,$$
where *k*_1_ is the rate constant for channel dimerization and *k*_−1_ the rate constant for channel dissociation (and τ = 1/*k*_−1_). To quantify the amphiphiles’ effect on the channel appearance rate (*f* = *k*_1_·[M]^2^, where [M] is the gA monomer concentration), we determined the channel appearance rates based on three (5–10 min) recordings obtained before and 10–20 min after addition of TX100, rTX100 or Cpsn (some results for Cpsn are from Ref. [24]). Only bilayers that did not break during amphiphile addition were used for the analysis.

The relative changes in the time-averaged channel “concentrations” were determined as the ratio of the product of the channel appearance rate and lifetime—measured before and after the amphiphile addition [98]:(8)$$\frac{{\left[D\right]}^{\mathrm{AM}}}{{\left[D\right]}^{\mathrm{cntl}}}=\frac{f^{\mathrm{AM}}\cdot \tau^{\mathrm{AM}}}{f^{\mathrm{cntl}}\cdot \tau^{\mathrm{cntl}}}=\frac{k_{1}^{\mathrm{AM}}\cdot {\left[M\right]}^{2}/k_{-1}^{\mathrm{AM}}}{k_{1}^{\mathrm{cntl}}\cdot {\left[M\right]}^{2}/k_{-1}^{\mathrm{AM}}}=\frac{K_{D}^{\mathrm{AM}}}{K_{D}^{\mathrm{cntl}}},$$
where the second and the third equalities hold in the limit [M]>>[D], such that [M]^AM^ ≈ [M]^cntl^. Changes in the free energy of gramicidin channel formation, $\Delta \Delta G_{\mathrm{tot}}^{0}$, which should be equal to the AM-induced change in $\Delta G_{\mathrm{def}}^{0}$, were evaluated as follows:(9)$$\Delta \Delta G_{\mathrm{tot}}^{0}\;(\approx \Delta \Delta G_{\mathrm{bil}}^{0})=-k_{B}T\cdot \ln\left\{\frac{K_{D}^{\mathrm{AM}}}{K_{D}^{\mathrm{cntl}}}\right\}=-k_{B}T\cdot \ln\left\{\frac{f^{\mathrm{AM}}\cdot \tau^{\mathrm{AM}}}{f^{\mathrm{cntl}}\cdot \tau^{\mathrm{cntl}}}\right\}.$$

The final results for a given experimental condition are reported as the mean ± standard deviation (SD) based on at least three independent measurements.

Though TX100 and Cpsn cause opposite changes in curvature [24], they both increase the channel appearance frequencies (*f*), lifetimes (τ) and, thus, the time-averaged channel densities in the bilayer (*f*·τ), and they reduce the bilayer deformation energy $\Delta \Delta G_{\mathrm{bilayer}}^{M\to D}$, with the larger effects on the shorter gA^−^(13) channels. Figure 9 shows results for TX100, rTX100 and Cpsn; Figure 10 (below) shows how all three amphiphiles increase gA channel lifetimes.

As was the case for Alm channels, one needs about three-fold higher [Cpsn] than [TX100] or [rTX100] to observe comparable changes in channel function, *f* and τ.

The results in Figure 9 can be interpreted further using the theory of elastic bilayer deformations, in which the bilayer deformation energy associated with the hydrophobic adaptation of the bilayer to an embedded protein is expressed as a biquadratic function in the channel-bilayer hydrophobic mismatch (*d*_0_ − *l*) and intrinsic lipid curvature (*c*_0_), [15], Equation (17):(10)$$\Delta G_{\mathrm{def}}=H_{B}\cdot {\left(d_{0}-l\right)}^{2}+H_{X}\cdot \left(d_{0}-l\right)\cdot c_{0}-H_{C}\cdot c_{0}^{2}$$
where *H*_B_, *H*_X_ and *H*_C_ are elastic coefficients that are functions of bilayer thickness, elastic moduli and channel radius (and include contributions from the energetic cost of redistributing the different components in a multi-component membrane, including the redistribution of the *n*-decane in our planar bilayer experiments). Equation (10) was derived from the theory of elastic bilayer deformations [15,99], and estimates for *H*_B_, *H*_X_ and *H*_C_ can be obtained from Equation (17) and Table 4 in reference [15]: *H*_B_ ≈ 350 kJ/(mol·nm^2^), *H*_X_ ≈ −290 kJ/mol and *H*_C_ ≈ −40 (kJ/mol)·nm^2^. The biquadratic form, however, applies more generally [6,14]. gA channels function, for example, is regulated by lipid bilayer thickness [58,59,100] and intrinsic lipid curvature [21,30], meaning that $\Delta G_{\mathrm{def}}^{0}$ can be expressed as a function of (*d*_0_ − *l*), *c*_0_ and maybe other terms. Using a Taylor expansion in (*l* − *d*_0_) and *c*_0_, $\Delta G_{\mathrm{def}}^{0}$ becomes [14]:(11)$$\begin{array}{ll} \Delta G_{\mathrm{def}}^{0}(d_{0}-l,c_{0}) & =\Delta G_{\mathrm{def}}^{0}(0,0)+\frac{\partial (\Delta G_{\mathrm{def}}^{0})}{\partial (d_{0}-l)}\cdot (d_{0}-l)+\frac{\partial (\Delta G_{\mathrm{def}}^{0})}{\partial c_{0}}\cdot c_{0}+\\ & +\frac{1}{2}\frac{\partial^{2}(\Delta G_{\mathrm{def}}^{0})}{\partial {\left(d_{0}-l\right)}^{2}}\cdot {\left(d_{0}-l\right)}^{2}+\frac{\partial^{2}(\Delta G_{\mathrm{def}}^{0})}{\partial (d_{0}-l)\partial c_{0}}\cdot (d_{0}-l)\cdot c_{0}+\frac{1}{2}\cdot \frac{\partial^{2}(\Delta G_{\mathrm{def}}^{0})}{\partial c_{0}^{2}}\cdot c_{0}^{2}+\ldots , \end{array}$$
where the first-order terms will be zero when the bilayer can be approximated as an elastic body. (The deformation energy for small decreases in (*d*_0_ − *l*) should equal that for small increases, with a similar argument holding for *c*_0_). The biquadratic form of $\Delta G_{\mathrm{def}}^{0}$ in Equation (10), thus, should be valid quite generally with the following:(12)$$H_{B}=\frac{1}{2}\cdot \frac{\partial^{2}(\Delta G_{\mathrm{def}}^{0})}{\partial {\left(d_{0}-l\right)}^{2}},\;H_{X}=\frac{\partial^{2}(\Delta G_{\mathrm{def}}^{0})}{\partial (d_{0}-l)\partial c_{0}},\;\mathrm{and}\;H_{C}=-\frac{1}{2}\cdot \frac{\partial^{2}(\Delta G_{\mathrm{def}}^{0})}{\partial c_{0}^{2}}.$$

In multi-component bilayers, the derivatives in Equations (11) and (12) include contributions from redistribution of bilayer components, whether it be lipids or hydrocarbon, meaning that Equation (10) should apply also for hydrocarbon-containing lipid bilayers, which can be approximated as elastic bodies [101,102,103,104]. The bilayer contribution to the gA channel monomer↔dimer equilibrium, $\Delta G_{\mathrm{bil}}^{M\to D}=\Delta G_{\mathrm{def}}^{D}-2\cdot \Delta G_{\mathrm{def}}^{M}$, can, thus, be expressed as $\Delta G_{\mathrm{bil}}^{M\to D}=H_{B}\cdot {\left(d_{0}-l\right)}^{2}+H_{X}\cdot \left(d_{0}-l\right)\cdot c_{0}$, where we assume that $\Delta G_{\mathrm{def}}^{M}=H_{C}\cdot c_{0}^{2}$.

The bilayer responds to the deformation by imposing a disjoining force (*F*_dis_) on the channel [6,8]:(13)$$F_{\mathrm{dis}}=-\frac{\partial (\Delta G_{\mathrm{def}}^{0})}{\partial (d_{0}-l)}=-2\cdot H_{B}\cdot \left(d_{0}-l\right)-H_{X}\cdot c_{0},$$
and changes in *F*_dis_—due, for example, to the adsorption of amphiphiles at the bilayer/solution interface—will be observable as changes in τ [8]:(14)$$\frac{\tau_{\mathrm{AM}}}{\tau_{\mathrm{cntl}}}=\exp\left\{-\frac{\left(F_{\mathrm{dis}}^{\mathrm{AM}}-F_{\mathrm{dis}}^{\mathrm{cntl}}\right)\cdot \delta }{k_{B}T}\right\},$$
where δ denotes the distance the two subunits move apart to reach the transition state for channel dissociation [60,100]. The lifetime changes, with the larger changes for the shorter gA^−^(13) channels with the larger hydrophobic mismatch, thus show that the amphiphiles alter the hydrophobic mismatch-dependent contribution to *F*_dis_, the $2\cdot H_{B}\cdot \left(d_{0}-l\right)$ term. The lipid bilayer thickness does not change [21,24,31], meaning that *d*_0_ − *l* is invariant, and we conclude that the amphiphiles alter *H*_B_ or the bilayer elasticity. The arguments can be strengthened by combining the gA^−^(13) and AgA(15) results, in which case Equation (14) becomes [8,14,21] the following:(15)$$\frac{\tau_{15}^{\mathrm{cntl}}\cdot \tau_{13}^{\mathrm{AM}}}{\tau_{15}^{\mathrm{AM}}\cdot \tau_{13}^{\mathrm{cntl}}}=\exp\left\{\frac{2\cdot (H_{B}^{\mathrm{AM}}-H_{B}^{\mathrm{cntl}})\cdot \delta \cdot (l_{13}-l_{15})}{k_{B}T}\right\},$$or
(16)$$\ln\{\tau_{13}^{\mathrm{AM}}/\tau_{13}^{\mathrm{cntl}}\}=\ln\{\tau_{15}^{\mathrm{AM}}/\tau_{15}^{\mathrm{cntl}}\}+\frac{2\cdot (H_{B}^{\mathrm{AM}}-H_{B}^{\mathrm{cntl}})\cdot \delta \cdot (l_{13}-l_{15})}{k_{B}T},$$leading to
(17)$$\frac{\tau_{13}^{\mathrm{amp}}}{\tau_{13}^{\mathrm{cntl}}}=\frac{\tau_{15}^{\mathrm{amp}}}{\tau_{15}^{\mathrm{cntl}}}\;\Rightarrow \;H_{B}^{\mathrm{amp}}=H_{B}^{\mathrm{cntl}},$$which holds irrespective of whether the amphiphiles (or changes in head group composition) alter lipid bilayer thickness.

We, therefore, examined the relationship between amphiphile-induced changes in $\tau_{13}^{\mathrm{AM}}/\tau_{13}^{\mathrm{AM}}$ and $\tau_{15}^{\mathrm{AM}}/\tau_{15}^{\mathrm{AM}}$, as shown in Figure 10.

As expected from previous studies [14,87,88,105], the amphiphiles produce greater changes in $\tau_{13}^{\mathrm{AM}}/\tau_{13}^{\mathrm{cntl}}$ than in $\tau_{15}^{\mathrm{AM}}/\tau_{15}^{\mathrm{cntl}}$, meaning that they increase bilayer elasticity.

The results in Figure 9 and Figure 10 show that amphiphile-induced changes in intrinsic monolayer curvature do not predict the changes in channel lifetime. This does not mean that intrinsic lipid curvature is not important. As previously shown, increasing the mole fraction of DOPE in DOPC/DOPE bilayers, which causes a negative change in *c*_0_ [29], leads to the same relative changes in the lifetimes of the shorter gA^−^(13) and the longer AgA(15) channels, Appendix A and [21], meaning they do not alter bilayer elasticity (Equation (17)).

### 2.3. Comparison of Amphiphile Effects on Alm and gA Channel Function

Alm and gA channels have different structures/organization (Figure 1) and would, therefore, be expected to respond differently to changes in lipid bilayer properties. Yet, if the amphiphile-induced changes in channel function reflect general changes in bilayer properties, the changes in Alm and gA channel function should be related. That is the case (Figure 11).

Figure 11A shows how the overall Alm channel activity, $\ln\left\{R_{\mathrm{Alm}}^{\mathrm{AM}}/R_{\mathrm{Alm}}^{\mathrm{cntl}}\right\}$, varies as a function of $\ln\{\tau_{15}^{\mathrm{AM}}/\tau_{15}^{\mathrm{cntl}}\}$, which is proportional to Δ*F*_dis_, the amphiphile-induced change in the disjoining force the bilayer imposes on the gA channels. Figure 11B shows how $\log\left\{\left(A_{\;2}^{\mathrm{AM}}/A_{\;1}^{\mathrm{AM}}\right)/\left(A_{\;2}^{\mathrm{cntl}}/A_{\;1}^{\mathrm{cntl}}\right)\right\}$, which is proportional to the change in the free energy between current levels 1 and 2 (cf. Equation (7)), varies as a function of $\ln\{\tau_{15}^{\mathrm{AM}}/\tau_{15}^{\mathrm{cntl}}\}$. Irrespective of the amphiphile-induced changes in curvature, the Alm channel activity and the relative stabilization of the higher current levels increase as the bilayer elasticity increases. The approximate linear relations between the changes in Alm and gA channel function show that both channels are regulated by changes in bilayer elasticity.

## 3. Discussion

Our results extend previous studies on the bilayer regulation of Alm channels [29,85] and gA channels [21,30], which showed that the distribution of current levels in Alm channels and the stability of gA channels are regulated by changes in intrinsic lipid curvature associated with changes in phospholipid head group composition that produce changes in intrinsic curvature, whether changes in head group bulk or electrostatic repulsion among the head groups. Positive changes in curvature promote the formation of gA channels but destabilize the high-conductance states in Alm channels; negative changes in curvature changes destabilize gA channels but promote the high-conductance states in Alm channels.

Changes in curvature, however, are not the only membrane property that is important for Alm and gA channel function. The changes produced by TX100, rTX100 and Cpsn, reversibly partitioning amphiphiles that produce positive (TX100 and rTX100) and negative (Cpsn) changes in curvature, differ qualitatively from the pattern observed with changes in head group composition. Irrespective of the changes in curvature, the amphiphiles promote the formation of gA channels, increase the stability of conducting Alm channels and shift the distribution of Alm channel current levels toward higher current levels.

If changes in intrinsic curvature were the only determinant of the changes in channel function, TX100 (and rTX100), on the one hand, and Cpsn, on the other, should produce opposite changes in Alm and gA channel function. The amphiphile’s dominant effects on channel function, therefore, must be due to other amphiphile-dependent effects, where changes in channel structure can be ruled because the structurally quite diverse amphiphiles produce little change in the magnitude of the current transitions in Alm, AgA(15) and the enantiomeric gA^−^(13) channels. Focusing on the gA^−^(13) and AgA(15) channels, the amphiphiles increase both lifetimes and channel formation frequency, with the larger changes in the gA^−^(13) as compared to AgA(15) channels (Figure 9 and Figure 10), which shows that their primary effect is to increase bilayer elasticity that results from the reversible partitioning of amphiphiles in lipid bilayers, e.g., [25,26,28]. A similar conclusion was reached by [27] based on studies on the effects of polyunsaturated fatty acids, known promoters of inverted hexagonal phases and negative curvature, e.g., [106].

Amphiphiles may, of course, alter other bilayer properties, including the surface potential associated with the partitioning of charged molecules at the interface [107,108] or the interfacial dipole potential associated with the partitioning of dipolar molecules into the interface [91]; see also [92]. Because Alm function would vary with changes in the electric field within the membrane, we added the amphiphiles to both sides of the membrane to minimize changes in the electric field across the membrane, cf. [93]. We also note that the Alm current levels changed little in the presence of the amphiphiles (Table 1).

Comparing the concentration-dependent effects of the amphiphiles, Cpsn is three-fold less potent than TX100 and rTX100 in producing changes in channel function, whether shifting the gA monomer↔dimer equilibrium to the right or promoting the formation of Alm channels and stabilizing the higher conduction levels in Alm channels. Analyzing the concentration-dependent shifts in the distribution among different Alm current levels (Figure 4 and Figure 5), we further found that the amphiphiles promoted higher current levels, as compared to the lower levels.

The Alm experiments were performed at nominal Alm concentrations ~10^−8^ M (in DOPC membranes, 10-fold higher in the DOPE-containing membranes). Given a partition coefficient of Alm into DOPC membranes, ~1.3 × 10^3^ [109], the Alm/lipid mole fraction in the membrane was ~10^−5^ (average distance between monomers ~250 nm, about two orders of magnitude longer than the decay length for elastic deformation [74,99]), far below the Alm/lipid mole fractions where Alm induces the formation of non-lamellar phases (~10^−2^ [110]), and Huang and colleagues demonstrated the transition from a surface-bound S state to a membrane-inserted I state (~10^−2^ [72]) or the mole fractions used to visualize Alm channels using X-ray scattering (10^−2^–10^−1^ [77,111]). Whereas the S→I transition described by Huang and colleagues (e.g., [38]) is likely to result from accumulated curvature stress associated with increasing surface density in the S state [72,74], the shift in the distribution of current transitions we observe probably do not result from changes in the curvature stress due to the S state because the mole fraction of Alm in the membranes is so low (average separation between monomers ~250 nm).

Our results do, however, provide insights into molecular features of Alm channels. First, we needed higher concentrations (mole fractions in the membrane) in the experiments with DOPC/DOPE mixtures. This may reflect the weaker binding of Alm to DOPC/DOPE membranes [73] as well as an increased insertion energy of the bilayer-spanning peptide due to Alm-bilayer hydrophobic mismatch [112]. Second, the opposite effects of curvature on Alm and gA channels suggest that the higher Alm channel conductance states have longer hydrophobic lengths than the lower conductance states. Knowing that *H*_X_ is negative (text below Equation (10)), the $H_{X}\cdot (d_{0}-l)\cdot c_{0}$ term in Equation (10) will decrease with increasing *l* when *c*_0_ < 0 (and increase with increasing *l* when *c*_0_ > 0). In the case of gA channels, where *l* < *d*_0_, the $H_{X}\cdot (d_{0}-l)\cdot c_{0}$ term will increase with decreasing *c*_0_, which likely accounts for the opposite effects of phospholipid head group-dependent changes in curvature on Alm and gA channels.

The similar effects of changes in elasticity on Alm and gA, as evident in Figure 11, likewise can be understood by considering how changes in bilayer elasticity alter Δ*G*_def_. Whenever *d*_0_ ≠ *l*, the $H_{B}\cdot {\left(d_{0}-l\right)}^{2}$ term in Equation (10) will be positive and, except for small hydrophobic mismatches, dominate. That is, Δ*G*_def_ will decrease with increasing bilayer elasticity (decreasing *H*_B_). When amphiphiles partition into bilayer/solution interfaces, they will alter both elasticity [14,24,25,26,28,87,88] and curvature, e.g., [24]. The amphiphile-induced changes in curvature, however, are modest compared with the changes observed when the head group composition is changed, as evident by comparing Figure 3 in [29], where *c*_0_ varied between −0.104 nm^−1^ in DOPC membranes and −0.468 nm^−1^ in DOPE membranes, with Figure 5 in [24], where *c*_0_ varied between −0.222 nm^−1^ at a TX100 mass fraction of 0.05 (mole fraction ≈ 0.08) and −0.258 nm^−1^ at a Cpsn mass fraction of 0.05 (mole fraction ≈ 0.14); *c*_0_ was −0.248 in the absence of amphiphile. The changes in head group composition produce a Δ*c*_0_ ≈ 0.364 nm^−1^, which should be compared to the amphiphile-induced changes, Δ*c*_0_ ≈ 0.036 nm^−1^, which provides a mechanistic basis for why the dominant effect of amphiphiles is the increased elasticity that results from their reversible partitioning into the bilayer/solution interface. The energetic consequences of the changes in curvature are modest because the changes in curvature are modest, which provides a mechanism for why amphiphiles have similar effects on Alm and gA channels and why amphiphile-induced changes in membrane protein function appear to be dominated by changes in bilayer elasticity [8,14,24].

## 4. Materials and Methods

### 4.1. Materials

1,2-Dioleoyl-*sn*-glycero-3-phosphocholine (DOPC), 1,2-dioleoyl-*sn*-glycero-3-phosphoethanolamine (DOPE) and 1,2-dioleoyl-*sn*-glycero-3-[phospho-L-serine] (DOPS) were from Avanti Polar Lipids (Alabaster, AL, USA) and used without further purification. *n*-Decane was 99.9% pure and from ChemSampCo (Trenton, NJ, USA). Alamethicin (Alm) from *Tricoderma viride* was from Sigma (St. Louis, MO, USA); it was used as supplied. The gramicidin A (gA) analog [Ala^1^]gA (AgA(15), with 15 amino acids in the sequence) and the sequence-shortened enantiomeric analog des-(D-Val^1^-Gly^2^)-gA^−^ (gA^−^(13), with 13 amino acids in the sequence) were synthesized and purified as described in [113]; they were 99+% pure, as determined by HPLC. Protein grade Triton X-100 (TX100) and reduced Triton X-100 (rTX100) were from Calbiochem-Novabiochem (La Jolla, CA, USA). Capsaicin (Cpsn) was from ICN Biomedicals (Aurora, OH, USA). Stock solutions of Alm, the gA analogs, TX100, rTX100 and Cpsn were prepared using dimethylsulfoxide (DMSO) (HPLC grade from Burdick & Jackson, Muskegon, MI, USA). The electrolyte solution was 0.1 or 1.0 M NaCl buffered to either pH 7.0 with 10 mM HEPES or pH 4.0 with 10 mM glycine–glycine (plus 1 mM EDTA) from Sigma.

### 4.2. Methods

Planar lipid bilayers were formed from 2.5% *w*/*v* suspensions of phospholipid in *n*-decane across a 1.5 mm hole in a Teflon^®^ partition separating the two electrolyte solutions, using the pipette method [114]; see also [115]. All experiments were performed at 25 ± 1 °C. Care was taken to minimize the total amount of lipid (and *n*-decane) that was added; the total volume of the lipid/*n*-decane solution usually was 1000-fold less than the volume of the aqueous solution.

For the Alm experiments, Alm was added to the *trans* side of the lipid bilayer; the *cis* side was the electrical ground. For the experiments with DOPC bilayers, we used ~10^−8^ M (nominal aqueous concentration, not corrected for adsorption of Alm to the bilayer and other surfaces in the chamber); for the experiments with DOPE or DOPS bilayers, we needed a 10-fold higher concentration to observe comparable channel activity. The applied potential was +150 mV.

For the gA experiments, we used the analogs AgA(15) and gA^−^(13), which allows for a test of how the effects of TX100, rTX100 or Cpsn depends on hydrophobic mismatch [13,24,98]; the applied potential was ±200 mV.

TX100, rTX100 or Cpsn were added to both aqueous solutions (both sides of the bilayer) during stirring as aliquots of 10 mM stock solutions in DMSO (stored at 5 °C). After amphiphile addition, the aqueous phases were stirred for at least 5 min before the measurements resumed. The total amount of added DMSO was less than 0.5% of the volume of the electrolyte solution, a concentration that has no effect on Alm or gA channel function.

The experimental protocol was as follows: after a stable membrane was formed, Alm was added to the *trans* side, or AgA(15) plus gA^−^(13) were added to both sides of the membrane, and control recordings were obtained. If channel activity was stable, the amphiphile was added by pipette and allowed to equilibrate before measurements resumed. If the membrane and channel activity were stable, another aliquot of the amphiphile was added and allowed to equilibrate, and then measurements resumed. If the membrane or the channel activity were unstable, the experiment was terminated.

Single-channel experiments were conducted using the bilayer-punch method [96] and a Dagan 3900A patch-clamp amplifier (Dagan Corp., Minneapolis, MN, USA) with a 3910 bilayer-expander module. The current signal in the Alm was filtered at 10 kHz, digitized at 20 kHz and digitally filtered at 8 kHz; the current signal in the gA experiments was filtered at 2 kHz, digitized at 20 kHz and digitally filtered at 500 Hz before the single-channel transitions were detected using transition-based algorithm [96] implemented in Visual Basic (Microsoft, Redmond, WA, USA).

## 5. Conclusions

Despite the different mechanisms of channel formation, the function of Alm and gA channels vary with changes in lipid bilayer composition and material properties (thickness, intrinsic curvature and the associated elastic moduli). Changes in head group composition, which alter curvature without altering elasticity, have opposite effects on Alm and gA channels. Amphiphiles, which alter elasticity with modest changes in curvature, have similar effects on Alm and gA channels, and the changes in the Alm channel function can be predicted from the changes in the gA channel function. This latter result is similar to what was found for other ion channels. That is, ion channels (and other membrane proteins) with different gating mechanisms respond similarly to amphiphile-induced changes in bilayer properties. The magnitude of the bilayer-mediated regulation of membrane protein function will differ among different proteins depending on their structure and the conformational transitions that underlie their function because the bilayer contribution to the free energy difference between different protein conformations is the difference in bilayer deformation energies association with the two conformations.

It is, in this context, important that any (indeed all) of the properties of lipid bilayers—thickness, curvature and their associated elastic moduli, etc.—collectively regulate protein function and that changes in one property are likely to be associated with changes in other properties. Again, this emphasizes the importance of the aggregate effects of these changes, e.g., in terms of the (amphiphile-induced) changes in bilayer deformation energy, which provides a mechanism for understanding the bilayer-mediated regulation of any membrane protein.

## Figures and Tables

**Figure 2 ijms-25-02758-f002:**
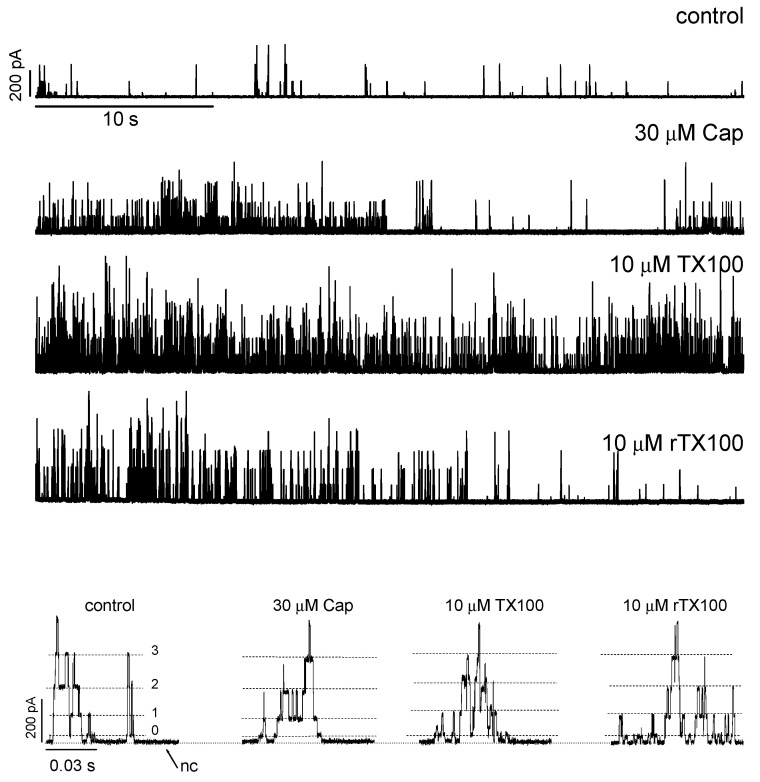
Amphiphile-induced changes in alamethicin channel activity. Cpsn, TX100 and rTX100 increase Alm channel activity. Top four records: 40 s recorded before the addition of amphiphile and after the addition of the indicated amphiphile (the control traces were similar for each amphiphile trace). The calibration bars in the top trace apply to all four traces. Bottom four traces show the effect of the amphiphiles at higher resolution; calibration bars in the control trace segment apply to all the trace segments. The stippled lines denote different current levels; they do not vary with amphiphile addition (Table 1) (DOPC, 1.0 M NaCl, pH 7.0, 150 mV).

**Figure 3 ijms-25-02758-f003:**
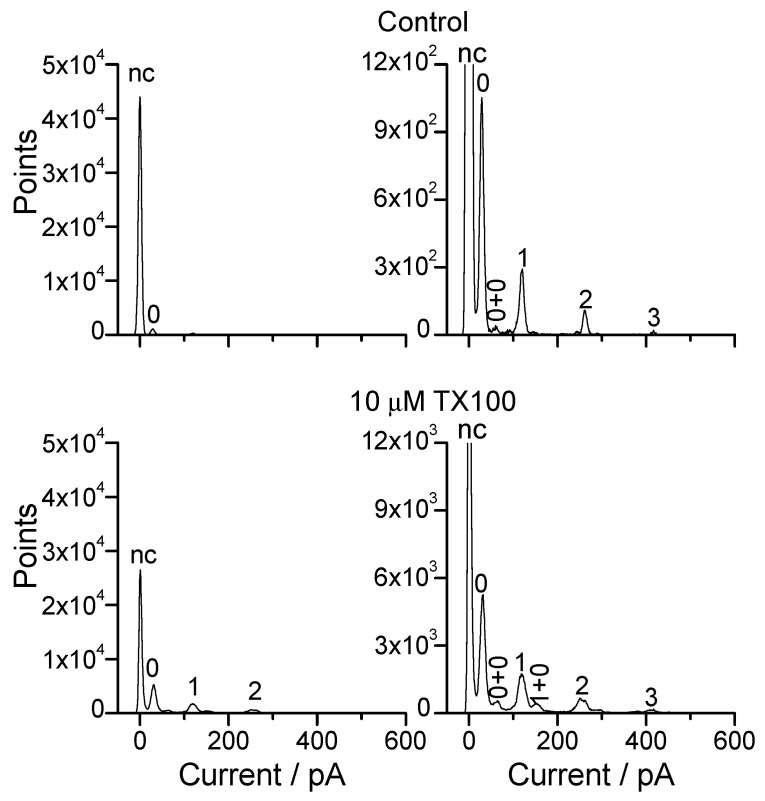
Current level (all-point) histograms showing the effects of TX100 on Alm channel function, results from one experiment. Top: results from a 40 s recording before the addition of TX100. Bottom: results from a 40 s recording in the same membrane a few min after the addition of 10 μM TX100. The right panels show the same results as the left but at an expanded scale for the ordinate. nc denotes the no-channel current level; the plots were aligned such that the nc peak is centered at 0 pA. The numbers over the peaks denote the identity of the channel state; two numbers indicate that the peak results from the superposition of two different channels (DOPC, 1.0 M NaCl, pH 7.0, 150 mV).

**Figure 4 ijms-25-02758-f004:**
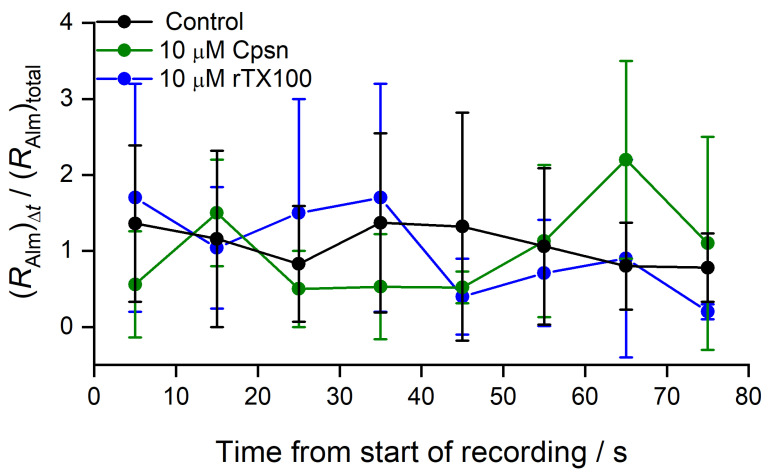
The variability of Alm channel activity as a function of time in the absence or presence of amphiphile. The ordinate denotes the channel activity, the time the channels reside in any conducting state relative to the no-channel state (*R*_Alm_, Equation (2)) over a 10 s time interval, normalized to the average activity over the total 80 s recording time. Mean ± S.D. based on at least three independent experiments at each condition (DOPC, 1.0 M NaCl, pH 7.0, 150 mV).

**Figure 5 ijms-25-02758-f005:**
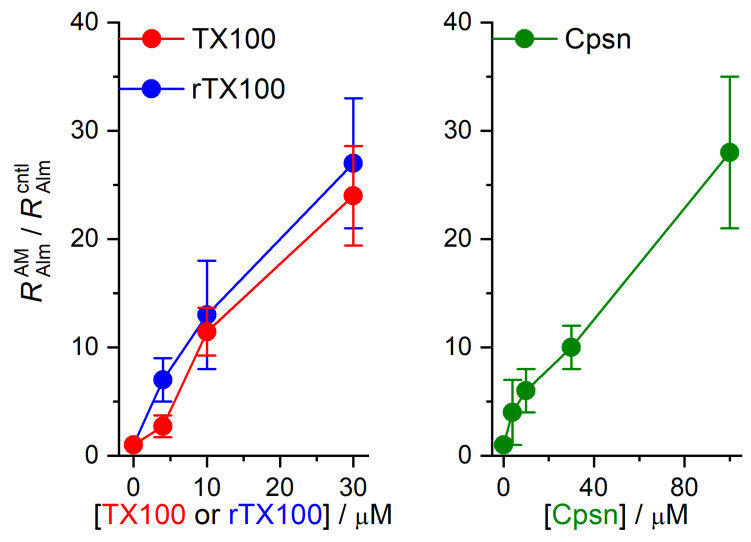
Effect of amphiphiles (TX100, rTX100 or Cpsn) on Alm channel activity. The ordinate displays the channel activity (Equation (2)) in the presence of amphiphile divided by the activity in the absence of amphiphile ($R_{\mathrm{Alm}}^{\mathrm{AM}}/R_{\mathrm{Alm}}^{\mathrm{cntl}}$, cf. Equation (3)). Mean ± S.D. based on at least three independent experiments, with one to three measurements, at each condition (DOPC, 1.0 M NaCl, pH 7.0, 150 mV).

**Figure 6 ijms-25-02758-f006:**
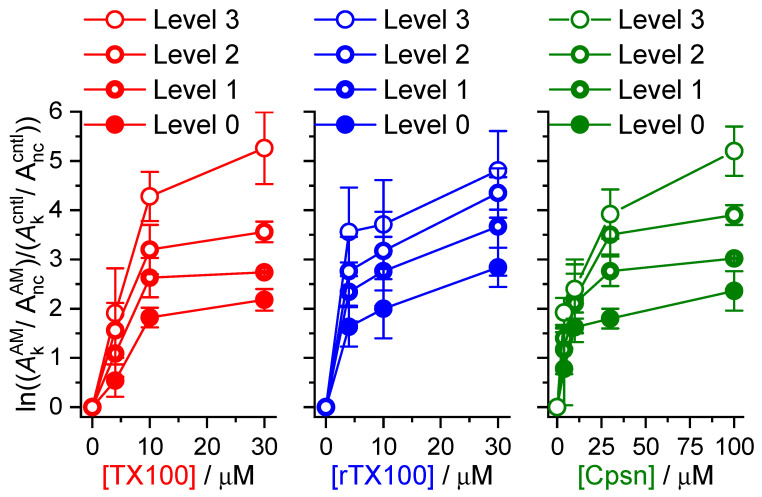
Effect of TX100, rTX100 or Cpsn on the distribution of Alm channel current levels relative to the nc level. The ordinate depicts the changes in $\ln\left\{\left(A_{k}^{\mathrm{AM}}/A_{\mathrm{nc}}^{\mathrm{AM}}\right)/\left(A_{k}^{\mathrm{cntl}}/A_{\mathrm{nc}}^{\mathrm{cntl}}\right)\right\}$, *k* = 0, 1, 2, 3, cf. Equation (6). Mean ± S.D. based on at least three independent experiments, each with one to three measurements, at each condition (DOPC, 1.0 M NaCl, pH 7.0, 150 mV).

**Figure 7 ijms-25-02758-f007:**
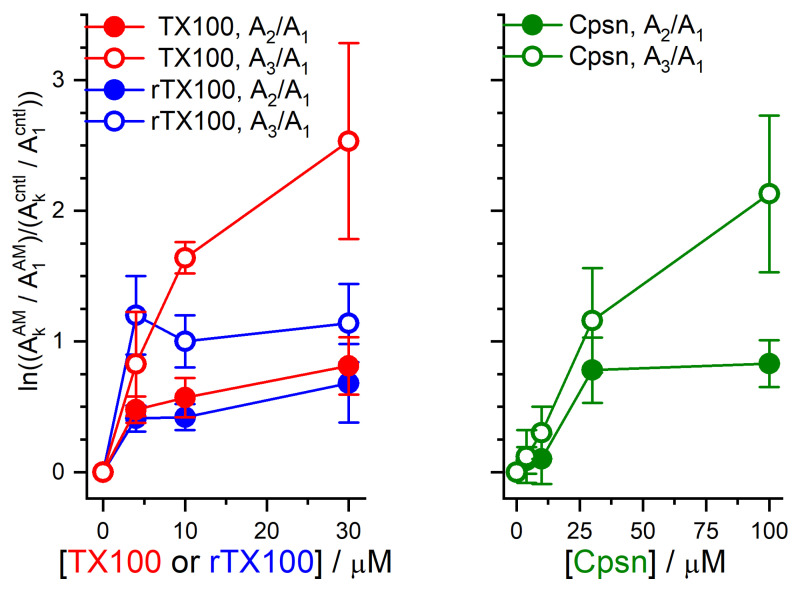
Effect of TX100, rTX100 or Cpsn on the distribution of time spent in different Alm current levels relative to the time spent in level 1. The ordinate shows ($\ln\left\{\left(A_{k}^{\mathrm{AM}}/A_{1}^{\mathrm{AM}}/(A_{k}^{\mathrm{cntl}}/A_{1}^{\mathrm{cntl}}\right)\right\}$, *k* = 2, 3cf. Equation (7)). Left, results for TX100 and rTX100. Right, results for Cpsn. Mean ± S.D. based on at least three independent experiments, each with one to three measurements, at each condition (DOPC, 1.0 M NaCl, pH 7.0, 150 mV).

**Figure 8 ijms-25-02758-f008:**
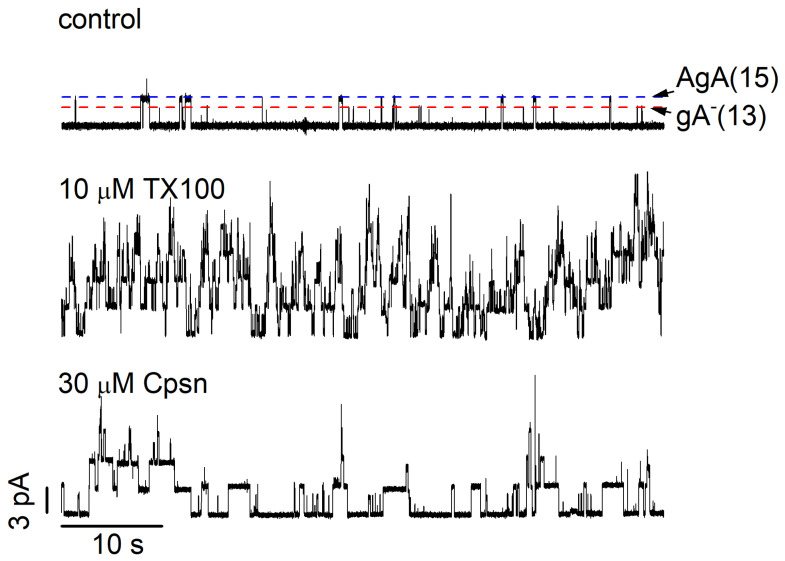
TX100 and Cpsn produce similar increases in gA channel activity. The three traces denote 60 s current traces recorded in the absence or presence of either 10 µM TX100 or 30 μM Cpsn (the control trace is from the TX100 experiment; similar single-channel activity was observed in the control trace for Cpsn). The experiments were performed using two different gA analogs, AgA(15) and gA^−^(13), which were added together to both sides of the bilayer. AgA(15) and gA^−^(13) channels can be distinguished by their current transition amplitudes (indicated by the horizontal dashed lines in the control current trace: blue for AgA(15) channels; red for gA^−^(13) channels). The calibration bars in the bottom trace apply to all traces (DOPC, 1.0 M NaCl, pH 7.0, 200 mV).

**Figure 9 ijms-25-02758-f009:**
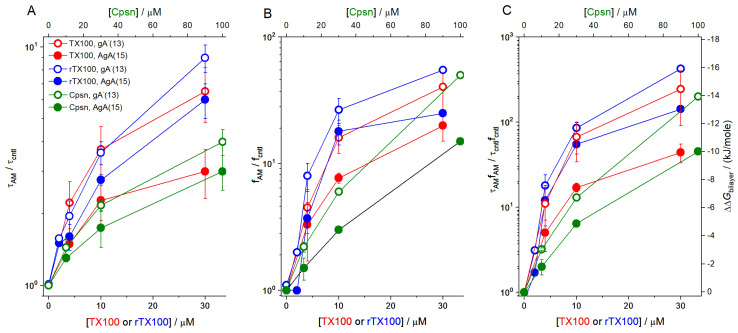
Effect of TX100, rTX100 and Cpsn on the lifetimes, appearance rates, channel activities and the change in the free energies of formation (Equation (9)) of AgA(15) and gA^−^(13) channels. (Panel (**A**)) shows results for τ_AM_/τ_cntl_; (panel (**B**)) shows results for *f*_AM_/*f*_cntl_; (panel (**C**)) shows results for τ_AM_·*f*_AM_/τ_cntl_·*f*_cntl_. To facilitate comparison of the results for the 13-residue and 15-residue channels, the results are displayed using logarithmic y axes. In the control experiments for TX100, τ_15_ and τ_13_ were 160 ± 13 ms and 11.6 ± 1.4 ms, respectively; in the rTX100 experiments, τ_15_ and τ_13_ were 131 ± 7 ms and 11.0 ± 0.4 ms, respectively; in the Cpsn experiments, τ_15_ and τ_13_ were 206 ± 14 ms and 15.5 ± 0.2 ms, respectively. Filled symbols—results for AgA(15) channels; open symbols—results for gA^−^(13) channels. Mean ± S.D. based on at least three independent experiments, each with three or more measurements, at each condition (DOPC, 1.0 M NaCl, pH 7.0, 200 mV).

**Figure 10 ijms-25-02758-f010:**
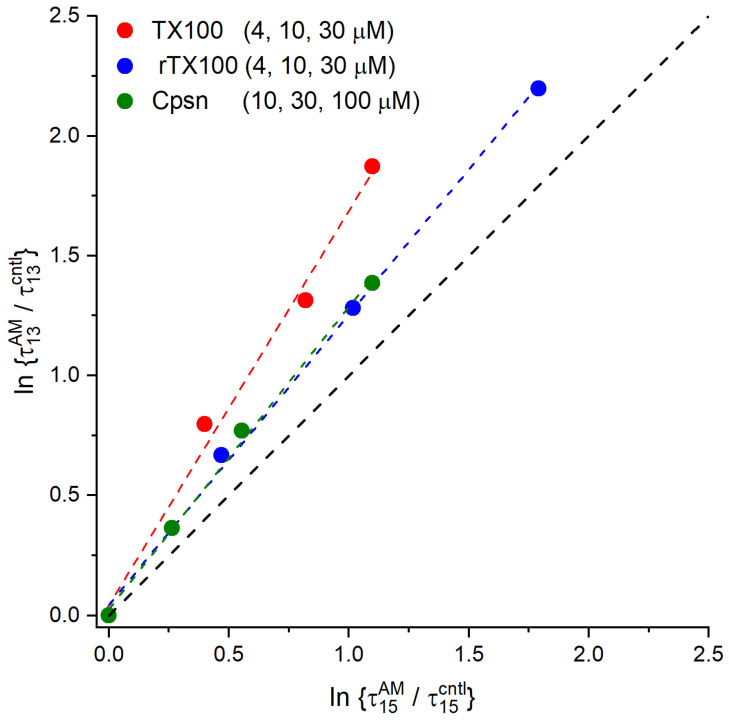
Amphiphiles produce larger relative changes in the lifetimes of gA^−^(13) channels, $\ln\{\tau_{13}^{\mathrm{AM}}/\;\tau_{13}^{\mathrm{cntl}}\}$, as compared to AgA(15) channels, $\ln\{\tau_{15}^{\mathrm{AM}}/\;\tau_{15}^{\mathrm{cntl}}\}$, based on results in Figure 9. The red, blue and green dashed lines denote linear fits to the result for TX100, rTX100 and Cpsn, respectively. For TX100, the slope was 1.64 ± 0.11, *r*^2^ = 0.986 (90% confidence interval for the slope, 1.29–1.99); for rTX100, the slope was 1.21 ± 0.04, *r*^2^ = 0.997 (90% confidence interval for the slope, 1.09–1.33); for Cpsn, the slope was 1.26 ± 0.05, *r*^2^ = 0.995 (90% confidence interval for the slope, 1.10–1.42). The black interrupted line has a slope of 1. (DOPC, 1.0 M NaCl, pH 7.0, 200 mV).

**Figure 11 ijms-25-02758-f011:**
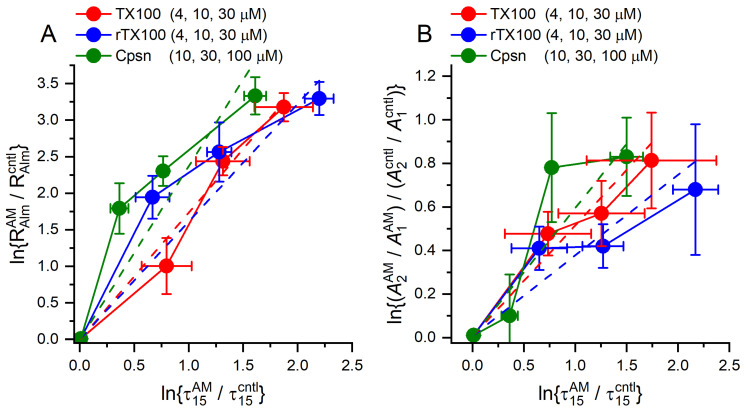
Amphiphile-induced changes in Alm function as functions of the changes in AgA(15) channel lifetimes. (**A**): Effect of TX100 (4, 10, 30 µM), rTX100 (4, 10, 30 µM) or Cpsn (10, 30, 100 µM) on Alm channel activity, expressed as $\ln\left\{R_{\mathrm{Alm}}^{\mathrm{AM}}/R_{\mathrm{Alm}}^{\mathrm{cntl}}\right\}$, cf. Equation (3), as functions of the corresponding changes in $\ln\left\{\tau_{15}^{\mathrm{AM}}/\tau_{15}^{\mathrm{cntl}}\right\}$. Based on results in Figure 5, Figure 7 and Figure 9. The red, blue and green dashed lines denote linear fits to the results, including 0 µM, for TX100, rTX100 and Cpsn, respectively. For TX100, the slope was 1.74, ± 0.08; *r*^2^ = 0.994 (90% confidence interval for the slope, 1.15–3.60); for rTX100, the slope was 1.61 ± 0.22, *r*^2^ = 0.940 (90% confidence interval for the slope, 0.89–2.33); for Cpsn, the slope was 2.37 ± 0.40, *r*^2^ = 0.920 (90% confidence interval for the slope, 1.15–3.60) (DOPC, 1.0 M NaCl, pH 7.0). (**B**): Effect of TX100, rTX100 or Cpsn on the distribution between Alm current level 1 and 2, expressed as $\ln\left\{\left(A_{\;2}^{\mathrm{AM}}/A_{\;1}^{\mathrm{AM}})/(A_{\;2}^{\mathrm{cntl}}/A_{\;1}^{\mathrm{cntl}}\right)\right\}$, cf. Equation (7), as functions of the corresponding changes in $\ln\left\{\tau_{15}^{\mathrm{AM}}/\tau_{15}^{\mathrm{cntl}}\right\}$. The dashed lines denote linear fits to the results, including 0 µM. For TX100, the slope was 0.51 ± 0.06, *r*^2^ = 0.959 (90% confidence interval for the slope, 0.33–0.70); for rTX100, the slope was 0.37 ± 0.08, *r*^2^ = 0872 (90% confidence interval for the slope, 0.13–0.61); for Cpsn, the slope was 0.59 ± 0.12, *r*^2^ = 0.920 (90% confidence interval for the slope, 0.24–0.95) (DOPC, 1.0 M NaCl, pH 7.0).

**Table 1 ijms-25-02758-t001:** Alm channel current levels 0–3 in the absence and presence of TX100, rTX100 and Cpsn.

	Level 0 (pA)	Level 1 (pA)	Level 2 (pA)	Level 3 (pA)
DOPC	4.5 ± 0.2	18.0 ± 0.5	38.4 ± 0.8	61 ± 1
+30 µM Cpsn	4.4 ± 0.3	17.7 ± 0.8	37.8 ± 0.9	60 ± 1
+30 µM TX100	4.5 ± 0.1	18.6 ± 0.6	39.5 ± 0.3	60 ± 1
+30 µM rTX100	4.5 ± 0.2	16.7 ± 0.6	36.8 ± 0.8	59 ± 1

Each datum (Mean ± S.D.) is based on current amplitude histograms (all-points histograms, cf. Figure 3) from at least three independent experiments conducted on different days, with one to three measurements for each condition on a given day (1.0 M NaCl, 10 mM HEPES (pH 7), 150 mV).

## Data Availability

The data presented in this study are available upon request from the corresponding author.

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
