# Peer review of "Intrinsic Lipid Curvature and Bilayer Elasticity as Regulators of Channel Function: A Comparative Single-Molecule Study"

_ijms, 2024, doi:10.3390/ijms25052758_

Round 1

Reviewer 1 Report

Comments and Suggestions for Authors

The manuscript is overall nicely written and data well presented. However, before further consideration, I'm recommending that authors provide an account on the minor issues described below:

-        Authors state that ‘The switch between the parallel (adsorbed) and  perpendicular (inserted) state can be understood in terms of a build-up of elastic curvature  stress in the bilayer [68], as the bilayer thickness decreases with increasing Alm mole-fraction until the mole-fraction where the switch from the adsorbed to inserted state occurs [62].’

It seems that no mention is made regarding the electric potential profile (either transmembrane or membrane dipole-potential generated) dramatic influence of triggering the transition between membrane adsorbed Alm molecules and inserted state. Authors are encouraged to read ‘Meet Me on the Other Side: Trans-Bilayer Modulation of a Model Voltage-Gated Ion Channel Activity by

Membrane Electrostatics Asymmetry’, PLoS ONE 6(9): e25276. doi:10.1371/journal.pone.0025276, demonstrating that the electric potential asymmetry on the membrane dipole potential does alters significantly the Alm activity.

-        As above, and regarding the ‘2.1. Amphiphiles modulate Alm channel function’ chapter of the manuscript, authors should note that certain amphiphiles addition on the side opposite to Alm addition, do also alter Alm activity, thus rendering the crucial effects of electric potential profile alteration, besides  changes in bilayer elasticity properties, on Alm (and possibly other ion-channel-like forming peptides) activity. (‘Meet Me on the Other Side: Trans-Bilayer Modulation of a Model Voltage-Gated Ion Channel Activity by Membrane Electrostatics Asymmetry’, PLoS ONE 6(9): e25276. doi:10.1371/journal.pone.0025276). For the sole benefit of the interested readers, I recommend that authors mention this effect as well.

-        On Fig. 4, authors quantify the probability of observing conducting Alm channels as a function of time in the absence or presence of amphiphile. However, by definition, probability cannot exceed 1, whereas numbers shown in the graph show differently, and this may confuse the readers (the same stands for Figs. 5 and 6 – probability cannot exceed 1). Authors are encouraged to clarify this aspect as well.

Author Response

We thank the reviewer for the comments, which have helped us improve the manuscript.

As a major change, we reorganized the presentation of the analysis, and incorporated it into the presentation of the results. We hope that this will minimize some of the reviewers’ problems with the presentation.

The reviewer is correct, we did not explicitly consider amphiphile-induced changes in the electric field in the membrane. We do not believe this to be a problem, however, because we added the amphiphiles symmetrically.

We added text, lines 215-220 and 753-760, to address this important point

The reviewer is correct that we did not use the word “probability” correctly.  Thank you for pointing this out.

We now describe explicitly that we focus on changes in channel activity. In the legend to Figure 4, we now write “The ordinate denotes the channel activity, the time the channels reside in any conducting state relative to the no-channel state (RAlm, Eq. ). We have made similar changes the legends to Figures 5 and 6.

Reviewer 2 Report

Comments and Suggestions for Authors

The manuscript "Intrinsic Lipid Curvature and Bilayer Elasticity as Regulators of Channel Function: A Comparative Single-Molecule Study" bu Ashrafuzzaman et al. studies how the amphiphiles Triton X-100 and capsaicin affect the function of alamethicin and gramicidin channels. This work builds upon this and other group's prior studies on the biophysics of membrane effects on ion channels. The main finding is that membrane elasticity is a more important regulator of channel function than curvature.

The work is of interest to the field of membrane biologists and biophysicists, and adds to knowledge about how membrane composition and properties affects protein function. I do have some questions about the work however.

1. The method for addition of Triton and capsaicin is to add it to both sides of the lipid bilayer. Wouldn't curvature of the membrane be better tested by adding the amphiphiles to only one side of the membrane?

2. The findings could be strengthened by using a second HII phase promoter, other than capsaicin, to confirm the conclusions. 

Author Response

We thank the reviewer for the comments, which have helped us improve the manuscript.

As a major change, we reorganized the presentation of the analysis, and incorporated it into the presentation of the results. We hope that this will minimize some of the reviewers’ problems with the presentation.

With respect to the question of curvature, we focused on the intrinsic curvature—the curvature an amphiphile would produce in a relaxed monolayer. If two such monolayers were to form a bilayer, the bilayer would be flat, but with a curvature frustration. The experiment the reviewer proposes is a good one, though difficult because amphiphiles tend to equilibrate between the two leaflets in a bilayer, but it goes beyond the scope of the present study.

We agree that it would be helpful with a second Hii phase promoter. We include a reference to experiments on gramicidin channels on lines 715-753, where polyunsaturated fatty acids, known promoters of Hii phases and negative curvature, increased bilayer elasticity with little effect of curvature. Polyunsaturated are difficult to work with, however, and we did not explore their effects on alamethicin channels.

Reviewer 3 Report

Comments and Suggestions for Authors

This study investigated the effects of amphiphilic compounds, known to affect the membrane curvature, on the activity of two model peptide ion channels, alamethicin and gramicidin. The effects were measured on single channel properties of these channels. Although Triton X-100 and r-Triton X-100 are known to have effects on membrane curvature that are opposite from those by Capsaicin, these compounds all increased the channel activity. It is concluded that the amphiphilic compounds affect the properties of these channels not by effects on the membrane curvature, but by changing the elasticity of the membranes.

Major points

1. In several parts of the study, conclusions are made on effects or based on comparisons between two or more conditions; the corresponding statistical tests were however not carried out.  They need to be provided. Thi concerns several of the main figures where comparisons are made, and especially the supplementary figures.

2. Some practical experimental information should be added in the text. This concerns the organization of the experiments. Were several conditions measured in each experiment, or only 1 control and 1 compound condition? Given that the control activity may vary between experiments, how were the compound conditions normalized to the control condition? Was there a perfusion in the measuring chambers, or were the solutions pipetted into it? How were drugs washed out?

3. In the second part of the discussion, from line 444 on, the contribution of different parts of equation 2 to the deformation energy are discussed. Further down, some possible values are indicated. However, the estimates and conclusions remain speculative and difficult to follow. The authors should try to fill in estimated values for the different terms, to better illustrate the relative contributions to the deltaG-def.

Specific points

1. Some of the text in the introduction (top of p.2) appears to be cut and pasted from the laboratory website.

2. In the context of the first determination of the conductances (Table 1) it would be instructive to show with a graph of an I-V curve how the conductance was determined.

3. In some graphs the meaning of the error bars is not indicated.

4. Legend to Fig. 5, "The probability of observing any current level relative to the no-channel level." In my opinion, this is not correct. What is shown in fact is the overall channel activity in the presence, divided by that measured in the absence of the compound.

5. In many figure legends it is referred to what is plotted on the y-axis, without specifying that the ln of this term is plotted. This should be corrected. On line 389, the log of a term is mentioned. This should probably be ln?

6. Supplemental Fig. S2, right traces, does the vertical calibration bar also apply to these traces? Please clarify, and provide calibration bar if the one on the left is not also valid for the right traces.

7. Fig. 8 shows on top a trace with the presence of both, AgA(15) and gA(13). Specify in the text whether the experiments with AM compounds were done in the simultaneous presence of both channel types, or only one. It seems that in the presence of both channel types, the analysis would be quite difficult.

8. In the first paragraph of the discussion, discussion of studies affecting membrane curvature, indicate which compounds were used.

9. Small typos:  line 95, "while" is hydrogen should be "white" is hydrogen; line 224, (..Figure 3A..), Figure 3A does not exist; line 619, "studies son the.." should be "studies on the..".

Author Response

We thank the reviewer for the detailed comments, which have helped us improve the manuscript.

As a major change, we reorganized the presentation of the analysis, and incorporated it into the presentation of the results. We hope that this will minimize some of the reviewers’ problems with the presentation.

With respect to statistics. Yes, we present many results, most of them do not require statistical analysis, but we have added information about the experimental basis for the reported data. The key information in the main text is in Figures 10 and 11, where we now provide 90% confidence limits for the regression lines. We also have updated the information in the Supporting Information, legends to Figures S3 and S4 and the associated text, including the likely changes in the spring constant.

We have added additional information about the experimental results on lines 886 – 892, as well as a reference to a JOVE video that shows in detail how the experiments were done. The video focuses on gramicidin channels, the extension to alamethicin is straightforward.

We agree that it would be helpful to have more information about the different terms that are discussed in the Discussion. We are working hard on filling this gap, but it is still a work-in-progress. Heeding the reviewer’s request we have added estimates for the three H coefficients below Eq. 10, which we refer to in the Discussion when we discuss the interplay between hydrophobic mismatch and curvature. For the remaining text, we would like to keep the text as is because we prefer to be vaguely right rather than exactly wrong (with due credit to Carver Reed).

Thank you for the specific points:

  1. With respect to the first point, there was no cutting and pasting, but there is only so many ways one can phrase this—and it needs to be there. We have modified the text in an attempt to overcome the problem.
  2. We calculated the conductance as cord conductances based on the current values measured at 150 mV. The conductance values have been converted back to currents.
  3. We have added this information in all figure legends.
  4. Thank you. We stand corrected and have revised the text here and elsewhere.
  5. Thank you, we have added this information where needed, and we have corrected the typo.
  6. Yes, the calibration applies ot both panels, which we note.
  7. Yes, the experiments were done in the presence of both, which we now note. We also outline briefly how the analysis was done. The key is to use current-transition amplitude histograms, where each channel type has its own well-defined peak. We add a reference.
  8. The first paragraph in the Discussion deals with changes in head group composition, we have added a sentence to that effect. The second paragraph deals with amphiphiles, and we believe we do mention what compounds that were used.
  9. The small typos have been corrected.

Round 2

Reviewer 1 Report

Comments and Suggestions for Authors

The authors have done a great deal of work toward improving the submission, which I'm gladly recommending for publication as is.

Only as minor suggestions, I recommend the following:

-        - The meaning of ‘A’ parameters in formulas 2 and 3, should be properly explained close to those expressions. Authors do indeed make it clear subsequently (in relation to formula 4) that those parameters relate to peak areas, but for the benefit of the reader the explanation should be given right on the spot, when they are being used first in the text.

-        -  In formula 4, it would be simpler I reckon if the nominator is denoted simply as the total area under the amplitude histogram.